# CONFPROBENCH: A CONFIDENCE EVALUATION BENCHMARK FOR MLLM-BASED PROCESS JUDGES

## ABSTRACT

Reasoning is the critical capability of multimodal large language models (MLLMs) to solve complex multimodal tasks, and judging the correctness of reasoning steps is crucial to improving this capability. Recently, MLLM-based process judges (MPJs) have been widely used to judge the correctness of reasoning steps in multimodal reasoning tasks. Therefore, evaluating the capability of MPJs is crucial for identifying their limitations and guiding future improvements. However, existing benchmarks for MPJs primarily focus on evaluating capabilities such as step correctness classification and reasoning process search, while overlooking a critical dimension: whether the confidence scores produced by MPJs at the step level are reliable. To fill this gap, we propose ConfProBench, the first comprehensive benchmark designed to systematically evaluate the reliability of step-level confidence scores generated by MPJs. This benchmark constructs three types of adversarially perturbed reasoning steps: Lexical Level, Syntactic Level, and Multimodal Level, to evaluate the robustness of MPJs' confidence under perturbations. Furthermore, we propose three novel evaluation metrics: Confidence Robustness Score (CRS), Confidence Sensitivity Score (CSS), and Confidence Calibration Score (CCS), which are designed to capture three complementary aspects of MPJs' confidence—robustness, sensitivity, and calibration. We evaluate 14 state-of-the-art MLLMs, including both proprietary and open-source models. Through extensive experiments, we reveal limitations in existing MPJs' confidence performance and provide competitive baselines, thereby paving the way for future research in this field. Our dataset is provided in the supplementary materials.

## 1 INTRODUCTION

Reasoning is a core capability of Multimodal Large Language Models (MLLMs) when tackling complex multimodal tasks Yan et al. (2024); Shi et al. (2024); Li et al. (2025); Xiang et al. (2024). Judging the correctness of each reasoning step is crucial for further enhancing this capability. As the reasoning chains generated by MLLMs become increasingly intricate, manually inspecting each intermediate step has become prohibitively costly. In response, recent studies have introduced MLLM-based Process Judges (MPJs) to assess step-by-step reasoning in multimodal tasks Pu et al. (2025); Chen et al. (2024); Huang et al. (2024); Sun et al. (2024); Zhang et al. (2024); Jiang et al. (2025). These MPJs analyze the reasoning process generated by MLLMs to identify potential flaws, improve interpretability, and facilitate targeted model improvements.

However, this paradigm shift raises a fundamental question: Can we trust the judgments made by MPJs? To address this, existing benchmarks evaluate multiple aspects of MPJs, such as step correctness, error type identification, and answer aggregation Ai et al. (2025); Xu et al. (2025); Wang et al. (2025). Nevertheless, they overlook an essential aspect: the reliability of the confidence scores produced by MPJs at the step level. Confidence not only reflects a model's self-assessed certainty but also directly affects controllability, reliability, and safety in downstream applications Geng et al. (2023). Under adversarial perturbations, robust and interpretable confidence scores are vital.

To fill this gap, we propose ConfProBench, the first benchmark specifically designed to systematically evaluate the confidence performance of MPJs. ConfProBench constructs perturbed variants of

reasoning steps using three types of adversarial perturbations: Lexical Level, Syntactic Level, and Multimodal Level. These perturbations support the assessment of confidence robustness.

Furthermore, we introduce a comprehensive evaluation metric suite that includes three core components: Confidence Robustness Score (CRS), Confidence Sensitivity Score (CSS), and Confidence Calibration Score (CCS). CRS measures the robustness of confidence under adversarial perturbations. CSS measures the sensitivity of confidence scores to erroneous reasoning steps. CCS evaluates the consistency between confidence scores and classification accuracy.

In summary, our main contributions are as follows:

- We propose ConfProBench, the first benchmark dedicated to systematically evaluating the confidence performance of MPJs, and the first benchmark to assess confidence robustness and sensitivity.
- We construct three types of adversarial perturbation data to evaluate the robustness of MPJs' confidence. We further introduce the first comprehensive confidence evaluation suite for MPJs, consisting of three complementary metrics: CRS, CSS, and CCS, which assess robustness, sensitivity, and calibration.
- We conduct comprehensive experiments on 14 state-of-the-art MPJs, including both proprietary and open-source models. Through fine-grained analysis using the core metrics and their subcomponents, we reveal critical limitations in current models' confidence performance and highlight directions for future improvement.

## 2 RELATED WORKS

### 2.1 CONFIDENCE EVALUATION AND ESTIMATION

Confidence is the estimated probability that a model's prediction matches the ground-truth label Guo et al. (2017). Assessing the confidence of large language models (LLMs) is essential for building reliable systems Geng et al. (2023). Most studies focus on calibration, which measures how well predicted confidence aligns with actual prediction accuracy Zhao et al. (2024); Geng et al. (2023). Confidence estimation and evaluation are distinct: the former extracts signals from the model, while the latter assesses their trustworthiness and stability Geng et al. (2023). Estimation methods include logit-based Duan et al. (2023), internal state-based Burns et al. (2022), consistency-based Manakul et al. (2023), and verbalized approaches Xiong et al. (2023). Verbalized methods prompt LLMs to express confidence via natural language or numerical values, and are valued for their model-agnostic design and efficiency Geng et al. (2023); Tian et al. (2023); Yang et al. (2024). We adopt this approach by prompting MPJs to produce step-level verbalized confidence and evaluate its robustness, sensitivity, and calibration.

| Benchmark | Multimodal | Step Annotation | MPJ-specific Confidence Metrics | Adversarial Perturbed Steps | Confidence Evaluation Paradigm |
|---|---|---|---|---|---|
| ProcessBench | No | Yes | No | No | No |
| PRMBench | No | Yes | Yes | No | No |
| VisualProcessBench | Yes | Yes | No | No | No |
| MPBench | Yes | Yes | No | No | No |
| ProJudgeBench | Yes | Yes | No | No | No |
| ConfProBench (Ours) | Yes | Yes | Yes | Yes | Yes |

Table 1: Comparison between related benchmarks with our ConfProBench.

### 2.2 BENCHMARKS FOR MLLM-BASED PROCESS JUDGES

In recent years, the process judgment capabilities of MLLMs have attracted increasing attention, and several related evaluation benchmarks have been proposed Wang et al. (2025); Xu et al. (2025); Ai et al. (2025). VisualProcessBench Wang et al. (2025) provides human-annotated step-wise correctness labels to evaluate the ability of multimodal Process Reward Models (PRMs) to identify erroneous steps in multimodal reasoning tasks. MPBench Xu et al. (2025) aims to assess the performance of multimodal PRMs across three tasks: determining the correctness of each reasoning

step (Step Correctness), selecting the optimal solution from multiple candidates (Answer Aggregation), and guiding the search of reasoning processes (Reasoning Process Search). ProJudgeBench Ai et al. (2025) is a multimodal, multidisciplinary benchmark specifically designed to evaluate the fine-grained error detection, classification, and diagnosis capabilities of MPJs.

Our ConfProBench is distinguished from prior benchmarks in three key aspects, as shown in Table 1. First, it is the first benchmark specifically designed for multimodal process judges (MPJs) with MPJ-specific, process-level confidence evaluation metrics, going beyond generic correctness or error-type assessment. Second, ConfProBench introduces three dimensions of adversarial perturbations—lexical, syntactic, and multimodal—providing a principled framework to evaluate the robustness of confidence under semantically preserving variations. Third, it enables comprehensive confidence evaluation through a suite of three complementary metrics (CRS, CSS, and CCS), which jointly capture robustness, sensitivity, and calibration at the step level, offering a finer-grained perspective than traditional confidence measures.

# 3 CONFPROBENCH

## 3.1 TASK DEFINITION

The multimodal process judging task in ConfProBench is framed as a binary classification problem. Our dataset contains two class labels: reasoning steps without errors are labeled as "correct" (1), while those with errors are labeled as "incorrect" (0). Specifically, the MPJ is required to output the probability that a reasoning step belongs to the correct class, which is used for both classification and confidence scoring.

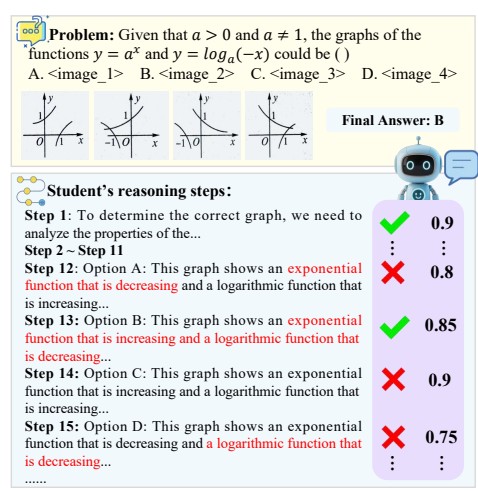

As illustrated in Figure 1, given a scientific problem $P$, its final answer $A$, and a step-by-step reasoning process $S = \{s_0, s_1, \cdots, s_{n-1}\}$ generated by a student model, the MPJ outputs a tuple $(l_i, p_i, e_i)$ for each reasoning step $s_i$. Here, $l_i \in \{1, 0\}$ indicates whether $s_i$ is belong to the correct class ($l_i = 1$) or incorrect class($l_i = 0$); $p_i \in [0, 1]$ denotes the probability that $s_i$ belongs to the correct class; and $e_i$ represents the error type if $s_i$ is belongs to the incorrect class.

The probability $p_i$ determines the predicted classification label and confidence score, while $l_i$ and $e_i$ assist in correcting potential inconsistencies in the result.

Figure 1: An example of the process judge task for MLLM-based process judges (MPJs), which perform binary classification of each reasoning step's correctness and provide associated confidence scores.

To obtain the binary step-level prediction, $p_i$ is converted into a correctness label $\hat{l}_i$ according to the following rule:

$$\hat{l}_i = \begin{cases} 1, & \text{if } p_i \geq 0.5, \\ 0, & \text{otherwise,} \end{cases} \tag{1}$$

Based on $\hat{l}_i$ and $p_i$, the confidence score $c_i$ is defined as:

$$c_i = \begin{cases} p_i, & \text{if } \hat{l}_i = 1, \\ 1 - p_i, & \text{if } \hat{l}_i = 0, \end{cases} \tag{2}$$

$p_i$, $\hat{l}_i$, and $c_i$ are subsequently used to compute the proposed evaluation metrics.

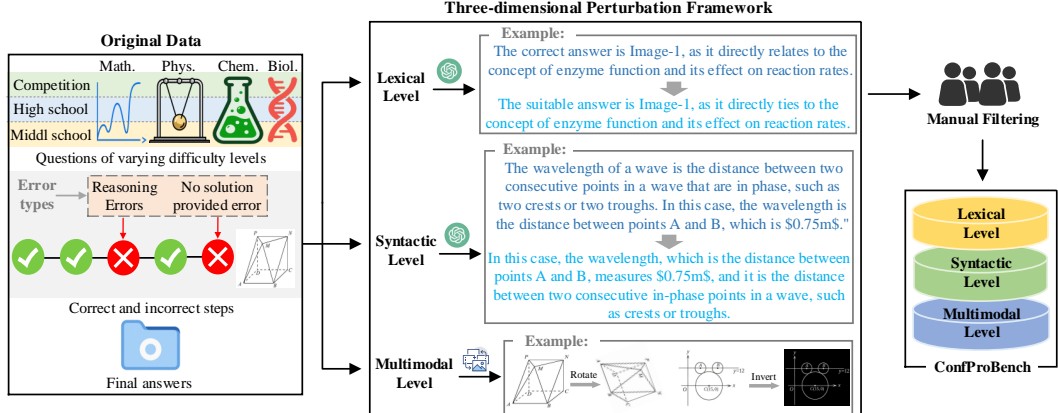

Figure 2: An overview of the data construction process for ConfProBench.

## 3.2 DATASET CONSTRUCTION

**Meta Data Extraction.** We construct our benchmark based on ProJudgeBench Ai et al. (2025) by sampling 1,200 problems spanning three difficulty levels (Middle School, High School and Competition), four scientific disciplines (Math, Physics, Chemistry, Biology), three modality types (Single Image, Multi Images, Pure Text), and seven types of reasoning errors (Numerical Calculation Error, Reasoning Error, Symbolic Calculation Error, Knowledge Error, Visual Interpretation Error, Question Understanding Error, and No Solution Provided). The resulting dataset maintains a balanced distribution across difficulty levels and scientific disciplines, offering a robust foundation for a comprehensive evaluation of MPJs' confidence performance. Please refer to Appendix B for detailed statistics of ConfProBench.

**Adversarial Perturbation Construction.** We design a three-dimensional perturbation framework spanning lexical, syntactic, and multimodal levels, which preserves semantics while diversifying expression. The framework is extensible to other perturbation types (e.g., numerical substitution, style rewriting, chart transformation). Multimodal perturbations apply only to Single-Image or Multi-Image samples. For balanced evaluation, we partition the 1,200 scientific problems into three equal subsets, each subjected to one perturbation type.

**Lexical Level**: We prompt GPT-4o to generate five distinct synonym-substituted versions for each reasoning step and randomly select one. In each version, at least one non-technical term, such as mathematical symbols, scientific terminology, programming syntax, technical jargon, or domain-specific abbreviations, is replaced with a semantically equivalent synonym. As many such terms as possible are substituted while ensuring grammatical correctness and semantic consistency.

**Syntactic Level**: We prompt GPT-4o to generate five distinct Syntactic Level versions for each step that preserve the original semantic information while exhibiting distinct syntactic structures, and randomly select one. Each Syntactic Level version strictly applies one of the following six predefined Syntactic Levels: (1) voice alternation (active to passive), (2) adverbial position adjustment, (3) clause order or structural variation, (4) phrase simplification or expansion, (5) inversion or emphasis construction, and (6) transformation of conditional, purposive, or resultative constructions.

**Multimodal Level**: We apply image-level perturbations to the image inputs of multimodal scientific problems. Specifically, one image transformation is randomly selected from the following set of operations: scaling, rotation, Gaussian noise injection, or color inversion. These transformations are designed to modify the low-level visual features of the input while preserving its semantic information.

Examples of each perturbation type are shown in Figure 2.

**Data Quality Control.** We conducted comprehensive manual verification of all adversarial perturbation results to ensure their quality and validity. Each reasoning step with Lexical Level was

examined to ensure that: (1) at least one non-technical term was replaced; (2) the original syntactic structure and semantic information were preserved; (3) technical terms and domain-specific vocabulary remained unchanged; (4) numerical values and mathematical expressions were not modified; and (5) the rewritten step was grammatically correct and fluent. Each syntactically transformed reasoning step was reviewed to ensure that: (1) no mathematical derivations, intermediate steps, or key expressions were omitted; (2) all numerical and symbolic content remained intact; (3) the sentence maintained its original meaning; and (4) the target structural transformation was appropriately applied. For Multimodal Levels, we examined each transformed image to ensure that the applied modifications did not introduce semantic information drift or obscure essential visual information. If a perturbed result failed to meet these criteria, we re-applied the corresponding perturbation procedure to the same reasoning step until a valid adversarial variant was obtained. Our verification process followed clear and objective standards, and the task required minimal subjectivity, inter-annotator agreement scores were not needed. Two PhD student from our team conducted the review and the rejection rate during this process was only 0.8%.

### 3.3 EVALUATION METRICS

Here we describe our evaluation framework in detail. As illustrated in Figure 3, it integrates robustness, sensitivity, and calibration aspects within a unified suite. To comprehensively evaluate the reliability of confidence scores produced by MPJs, we introduce a multi-dimensional suite of evaluation metrics, as illustrated in Figure 3. This metric suite is designed to capture three complementary aspects of confidence performance: robustness, sensitivity, and calibration. These three metrics form a comprehensive framework to assess whether an MPJ can reliably express the uncertainty of its predictions, which is an essential capability for trustworthy MPJs.

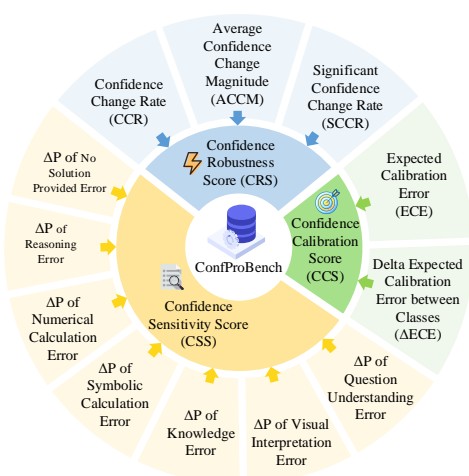

Figure 3: An overview of the proposed evaluation metric suite, consisting of three core metrics: CRS, CSS, and CCS. Each core metric is composed of a set of sub-metrics.

**Confidence Robustness Score (CRS).** We define the Confidence Robustness Score (CRS) to measure the robustness of confidence under designed adversarial perturbations, including Lexical Level, Syntactic Level, and Multimodal Level. Since these perturbations preserve the semantic consistency of the reasoning steps, an ideal process judge should maintain consistent confidence scores across both perturbed and unperturbed inputs.

CRS integrates three sub-metrics to quantify confidence robustness. Let $c_i$ represent the original confidence score, and $c'_i$ represent the confidence score after perturbation. For each pair of original confidence score and post-perturbation confidence score, we compute the following sub-metrics:

(1) Confidence Change Rate (CCR): The proportion of reasoning steps in which the confidence scores change after perturbation. Specifically, if the absolute difference in confidence exceeds a small threshold $\epsilon$, we consider the confidence to have changed. CCR is defined as:

$$\text{CCR} = \frac{1}{N} \sum_{i=1}^{N} \mathbb{I}\left(|c_i - c'_i| > \epsilon\right), \tag{3}$$

Where $N$ is the total number of reasoning steps, and $\mathbb{I}(\cdot)$ is the indicator function, which is used to check if a condition is met. It returns 1 if the condition is true, and 0 if it is false. A lower CCR indicates greater robustness of confidence.

| Model | CRS↑ | CSS↑ | CCS↑ | Avg.↑ |
|---|---|---|---|---|
| **Open-source MLLMs** | | | | |
| InternVL3-8B | 77.41 | 11.55 | 25.97 | 38.31 |
| InternVL3-14B | 50.78 | 21.19 | **46.75** | 39.57 |
| InternVL3-38B | 49.92 | **30.62** | 44.49 | 41.68 |
| MiniCPM-V-2_6 | 68.05 | 6.60 | -47.95 | 8.90 |
| Qwen2.5-VL-3B | 74.71 | 3.15 | 2.73 | 26.86 |
| Qwen2.5-VL-7B | 71.19 | 10.38 | 15.80 | 32.46 |
| Qwen2.5-VL-32B | **81.06** | 15.93 | 41.60 | **46.20** |
| Qwen2.5-VL-72B | 77.45 | 19.93 | 25.30 | 40.89 |
| QVQ | 74.17 | 12.60 | 30.69 | 39.15 |
| **Proprietary MLLMs** | | | | |
| GPT-4o | 57.37 | 30.71 | **62.00** | 50.03 |
| GPT-4o-Mini | 65.58 | 13.03 | 47.73 | 42.11 |
| GPT-4.1 | 73.62 | 38.51 | 37.65 | 49.93 |
| Gemini-2.5-flash | 63.08 | 48.29 | 48.62 | 53.33 |
| Gemini-2.5-flash-nothinking | 51.20 | 42.13 | 51.55 | 48.29 |
| Gemini-2.5-Pro | **76.90** | **57.73** | 44.88 | **59.84** |
| GPT-5 | 64.27 | 51.59 | 55.38 | 57.08 |

Table 2: The main results across different MLLM-based Process Judges (MPJs) on ConfProBench. The best performance for each metric is shown in bold, while the second-best is underlined.

(2) Average Confidence Change Magnitude (ACCM): The average magnitude of confidence change across all steps where the change exceeds the small threshold $\epsilon$. Specifically, we define:

$$\text{ACCM} = \frac{1}{|S|} \sum_{i \in S} |c_i - c_i'|, \tag{4}$$

$$\text{where } S = \{i \mid |c_i - c_i'| > \epsilon\},$$

A smaller ACCM indicates greater robustness of confidence.

(3) Significant Confidence Change Rate (SCCR): It refers to the proportion of reasoning steps where the confidence score changes beyond a predefined threshold $\delta$. We refer to this threshold as the significant threshold, which is set to $0.2$ in our experiments. This parameter can be adjusted according to different application needs. The formal definition of SCCR is as follows:

$$\text{SCCR} = \frac{1}{N} \sum_{i=1}^{N} \mathbb{I}\left(|c_i - c_i'| > \delta\right), \tag{5}$$

A lower SCCR indicates greater robustness of confidence.

We combine the three sub-metrics above to define the CRS as follows:

$$\begin{aligned} \text{CRS} = w_1 \cdot (1 - \text{CCR}) + w_2 \cdot (1 - s \cdot \text{ACCM}) \\ + w_3 \cdot (1 - s \cdot \text{SCCR}), \end{aligned} \tag{6}$$

where $w_1$, $w_2$, and $w_3$ are the weights of the three sub-metrics, and $s$ is a scaling factor.

**Confidence Sensitivity Score (CSS).** We propose Confidence Sensitivity Score (CSS), a novel metric that quantifies how sensitively confidence scores respond to reasoning errors.

For each error type $t \in \mathcal{T}$, let $\overline{p}_t$ denote the average value of $p_i$ over all steps labeled with the ground-truth error type $t$, and let $\overline{p}_{\text{correct}}$ denote the average $p_i$ over all steps labeled as ground-truth correct. We then define $\Delta p_t$ as the difference between $\overline{p}_{\text{correct}}$ and $\overline{p}_t$, as follows:

$$\Delta p_t = \overline{p}_{\text{correct}} - \overline{p}_t, \tag{7}$$

A larger $\Delta p_t$ indicates that $p_i$ significantly decreases when encountering an error of type $t$, showing that $p_i$ is sensitive to this type of error. Conversely, a smaller or even negative $\Delta p_t$ suggests that

$p_i$ has weak or no ability to recognize this error type. Since $c_i$ is derived from $p_i$ through a simple linear transformation, the sensitivity of $p_i$ to reasoning errors directly reflects the model confidence's sensitivity.

To assess the overall confidence sensitivity, we define CSS as the average of $\Delta p_t$ across all error types:

$$\text{CSS} = \frac{1}{|\mathcal{T}|} \sum_{t \in \mathcal{T}} \Delta p_t, \tag{8}$$

where $\mathcal{T}$ is the set of all non-empty error types in the dataset.

**Confidence Calibration Score (CCS).** Confidence Calibration Score (CCS) evaluates the consistency between the confidence score and the actual accuracy of predictions . It incorporates two aspects of calibration errors: the overall Expected Calibration Error (ECE) Guo et al. (2017), and the gap in ECE between classes, denoted as $\Delta\text{ECE}_{\text{cls}}$.

The ECE is defined as:

$$\text{ECE} = \sum_{m=1}^{M} \frac{|B_m|}{n} \cdot |\text{acc}(B_m) - \text{conf}(B_m)|, \tag{9}$$

where $B_m$ is the $m$-th bin obtained by equally dividing the confidence range into $M$ intervals, $|B_m|$ denotes the number of samples in bin $B_m$, and $n$ is the total number of samples. $\text{acc}(B_m)$ and $\text{conf}(B_m)$ represent the average accuracy and average confidence of samples within bin $B_m$, respectively.

To better capture class-specific calibration performance, we compute the ECE separately for the correct and incorrect categories of reasoning steps, denoted as: $\text{ECE}_{\text{correct}}$ and $\text{ECE}_{\text{incorrect}}$. The class-wise calibration gap is then defined as

$$\Delta\text{ECE}_{\text{cls}} = |\text{ECE}_{\text{correct}} - \text{ECE}_{\text{incorrect}}|, \tag{10}$$

A smaller $\Delta\text{ECE}_{\text{cls}}$ indicates more balanced confidence calibration performance across different classes.

Combining ECE and $\Delta\text{ECE}_{\text{cls}}$, we define CCS as follows:

$$\text{CCS} = 0.5 \cdot (1 - s \cdot \text{ECE}) + 0.5 \cdot (1 - \Delta\text{ECE}_{\text{cls}}), \tag{11}$$

Where $s$ (set to 5) is a scaling factor.

Our proposed CRS, CSS, and CCS go beyond classical metrics like ECE Guo et al. (2017) by capturing step-level robustness, sensitivity, and calibration under semantic-preserving perturbations. Such fine-grained step-level evaluation is particularly important for multimodal process judges to ensure reliable and interpretable reasoning.

All three metrics—CRS, CSS, and CCS—employ parameter choices guided by theoretical and practical considerations. For CRS, the thresholds $\epsilon$ and $\delta$ distinguish meaningful confidence changes from minor fluctuations, while the scaling factor $s$ ensures that sub-metrics wites contribute comparably to the overall score; the weights $w_1$, $w_2$, and $w_3$ balance emphasis across sub-metrics. For CSS, $\Delta p_t$ is naturally bounded within [-1,1], and averaging across all error types provides a balanced, step-level sensitivity measure that treats different error types equally. For CCS, the scaling factor $s$ amplifies the impact of the typically smaller ECE relative to the class-wise calibration gap $\Delta\text{ECE}_{\text{cls}}$, and equal weighting between these components captures both global calibration and class-specific fairness. Across all metrics, these parameter settings are designed to suppress noise, balance contributions from different sub-components, and maintain fine-grained interpretability of step-level reasoning, making them particularly suitable for multimodal process judges. Accordingly, for both CRS and CCS, each sub-metric is subtracted from 1 so that higher values indicate stronger confidence robustness and better confidence calibration, respectively.

# 4 EXPERIMENTS

## 4.1 EXPERIMENTAL SETTINGS

To provide a comprehensive evaluation on ConfProBench, we assess both proprietary and open-source MPJs. The proprietary MPJs include GPT-4o OpenAI (2024b), GPT-4o-Mini OpenAI (2024c), GPT-4.1 OpenAI (2024a), Gemini-2.5-flash (Dynamic thinking) DeepMind (2025a), and Gemini-2.5-flash-nothinking DeepMind (2025b). The open-source MPJs span a variety of architectures and parameter scales, including InternVL3 (8B, 14B, 38B) Zhu et al. (2025), Qwen2.5-VL (3B, 7B, 32B, 72B) Bai et al. (2025), MiniCPM-V-2_6 (8B) Yao et al. (2024), and QVQ (72B) Qwen Team (2024).

For reproducibility and transparency, detailed parameter settings, including thresholds, scaling factors, and weights for CRS, CSS, and CCS, are provided in the Appendix C. All MPJs use a unified prompt template, with detailed prompt designs provided in Appendix D. All metric values are presented as percentages in the tables.

To enable a consistent and fair comparison between proprietary and open-source MPJs, we adopt verbalized confidence as the evaluation signal. This choice is motivated by the fact that proprietary models typically do not provide access to logits or internal probability distributions, making verbalized confidence the feasible and generally comparable signal across all models (Xiong et al., 2023).

## 4.2 RESULTS AND ANALYSIS

The primary experimental results for the three core metrics CRS, CSS, and CCS are presented in Table 2. To enable more fine-grained analysis, the results of the sub-metrics that constitute these core metrics are reported separately in Tables 4–6 in appendix.

## 4.3 RESULTS AND ANALYSIS

The primary experimental results for the three core metrics—CRS, CSS, and CCS—are presented in Table 2. To support more fine-grained analysis, the decomposition results of the sub-metrics that constitute these core metrics are reported separately in Tables 4–6 in the appendix. With the inclusion of the newly released **Gemini-2.5-Pro** and **GPT-5**, several best and second-best results are updated accordingly.

**Confidence Robustness Analysis.** As shown in Table 2, **Gemini-2.5-Pro** achieves the highest CRS score (76.90) among all proprietary MPJs, surpassing previous models such as GPT-4.1 (73.62) and Gemini-2.5-flash (63.08). However, several open-source MPJs—including InternVL3-8B (77.41), Qwen2.5-VL-32B (81.06), Qwen2.5-VL-72B (77.45), and QVQ (74.17)—still outperform proprietary MPJs on CRS, indicating that confidence robustness does not simply scale with model size or proprietary tuning. This further highlights the effectiveness of the CRS metric in revealing robustness gaps. Even the strongest MPJs remain far below the theoretical maximum, suggesting substantial room for improvement.

Table 4 provides additional insights through CRS sub-metrics. For example, Qwen2.5-VL-32B exhibits low CCR, ACCM, and SCCR, indicating that confidence changes are infrequent, mild, and seldom exceed the significance threshold. In contrast, InternVL3-38B demonstrates high values on all three sub-metrics, indicating frequent and substantial confidence fluctuations under perturbations, resulting in its low CRS.

**Confidence Sensitivity Analysis.** Gemini-2.5-Pro =achieves the **highest CSS score (57.73)**, outperforming all existing proprietary MPJs, including Gemini-2.5-flash (48.29). **GPT-5** becomes the **second-best** model in CSS (51.59), reshaping the sensitivity ranking among proprietary models. Although these new models demonstrate noticeable improvement, the CSS scores remain far from the theoretical upper bound, suggesting room for further enhancement.

Table 5 shows that proprietary MPJs generally achieve higher average confidence changes ($\Delta p$) across error types. In contrast, some open-source MPJs, such as Qwen2.5-VL-3B ($-4.22$) and

MiniCPM-V-2.6 ($-21.62$), display negative $\Delta p$ on QUE, indicating unreliable confidence that does not properly reflect actual reasoning errors.

**Confidence Calibration Analysis.**   As shown in Table 2, proprietary MPJs significantly outperform open-source MPJs in CCS. Among them, GPT-4o achieves the highest CCS score of 62.00, indicating substantially stronger confidence calibration performance than other MPJs. However, this is still far from the theoretical upper bound, suggesting ample room for further improvement. In contrast, open-source MPJs, such as MiniCPM-V-2.6 and Qwen2.5-VL-3B, perform relatively poorly. Notably, MiniCPM-V-2.6 exhibits a negative CCS ($-47.95$), indicating suboptimal confidence calibration performance, involving both ECE and $\Delta$ECE. Analysis of Table 5 shows that this negative CCS is primarily due to a high ECE of 45.16, suggesting a significant mismatch between the model's predicted confidence and the actual correctness across many reasoning steps.

Furthermore, as shown in Table 6, it can be observed that across all MPJs, $ECE_{correct}$ is consistently much lower than $ECE_{incorrect}$, resulting in relatively large $\Delta$ECE values. This indicates an imbalance in confidence calibration across classes. Therefore, the calibration performance on erroneous reasoning steps remains unsatisfactory and calls for urgent improvement.

**Average Score Comparison.**   Gemini-2.5-Pro achieves the highest average score (59.84), establishing a new state-of-the-art among all MPJs. GPT-5 ranks second (57.08), surpassing GPT-4o (50.03), GPT-4.1 (49.93), and Gemini-2.5-flash (53.33). Proprietary MPJs thus occupy the top positions, reflecting the benefits of advanced training and alignment techniques.

Most open-source MPJs remain within the 30–40 range, with MiniCPM-V-2.6 scoring the lowest (8.90) primarily due to poor calibration performance. The InternVL series continues to outperform the Qwen2.5-VL series, and its performance scales positively with model size, with InternVL3-38B achieving the best average score among open-source models (41.68). Despite improvements in larger Qwen models from 3B to 32B, performance degrades at 72B, indicating that scaling alone does not guarantee improved confidence quality.

**Impact of Model Scale on Confidence Performance.**   As shown in Table 2, model scale exhibits varying effects on different aspects of confidence performance. Specifically, no clear positive correlation is observed between model size and confidence robustness. For example, within the InternVL3 series, CRS consistently decreases as model size increases from 8B to 38B. In contrast, confidence sensitivity generally improves with scale. For instance, in the Qwen2.5-VL series, CSS rises from 3.15 (3B) to 19.93 (72B), indicating enhanced confidence sensitivity. As for calibration, larger models tend to perform better. For example, Qwen2.5-VL's CCS increases from 2.73 (3B) to 41.60 (32B), but drops again at 72B, suggesting that increasing model size alone does not ensure better calibration.

**Impact of Thinking Mode on Confidence Performance.**   Table 2 presents the core metric results for Gemini-2.5-flash and its no-thinking variant. Results show that enabling the thinking process enhances confidence robustness under input perturbations, as evidenced by a higher CRS. Additionally, Gemini-2.5-flash exhibits a 6.16-point improvement in CSS, suggesting that the thinking process enhances the model's sensitivity to erroneous reasoning steps. However, its CCS is lower than that of the no-thinking variant, indicating that the thinking process does not necessarily improve confidence calibration quality.

## 5 CONCLUSION

We present ConfProBench, the first benchmark for evaluating the reliability of step-level confidence scores produced by MPJs. It introduces three types of adversarial perturbations to assess the robustness of MPJs' confidence under input variations. Furthermore, it proposes a comprehensive evaluation suite comprising three complementary metrics: CRS, CSS, and CCS, which measure the robustness, sensitivity, and calibration of MPJs' confidence. Extensive experiments reveal key limitations in current MPJs' confidence performance and establish strong baselines, paving the way for future research in this area. Beyond these contributions, we suggest two future directions. First, conducting human confidence annotations and introducing new consistency metrics to assess the

alignment between MPJ confidence and expert judgments. Second, extending ConfProBench to encompass safety-critical scenarios where highly reliable confidence estimation is essential.

## 6 ETHICS STATEMENT

This research does not involve human subjects, personally identifiable information, or sensitive data; therefore, no Institutional Review Board (IRB) approval was required. All datasets used are publicly available and released under appropriate licenses. Our work poses no apparent ethical risks and has no conflicts of interest.

## 7 REPRODUCIBILITY STATEMENT

To ensure reproducibility, we provide detailed descriptions of the models and experimental settings in the main text, with additional hyper-parameter configurations and implementation details included in the appendix. All datasets used in our experiments are publicly available. Furthermore, we provide the source code, configuration files, and execution scripts in the supplementary material, enabling other researchers to faithfully reproduce our results.

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

## A  THE USE OF LARGE LANGUAGE MODELS (LLMS)

We employ Large Language Models (LLMs) for grammar checking in our paper.

## B  DETAILED STATISTICS OF CONFPROBENCH

The detailed statistics of ConfProBench are summarized in Table 3.

## C  PARAMETER SETTINGS AND THEORETICAL RANGES FOR EVALUATION METRICS

This appendix provides the detailed parameter settings and theoretical ranges for the three evaluation metrics—Confidence Robustness Score (CRS), Confidence Sensitivity Score (CSS), and Confidence Calibration Score (CCS)—used in ConfProBench.

### C.1  CONFIDENCE ROBUSTNESS SCORE (CRS)

CRS measures the robustness of confidence under semantic-preserving adversarial perturbations (Lexical, Syntactic, Multimodal). It consists of three sub-metrics: Confidence Change Rate (CCR), Average Confidence Change Magnitude (ACCM), and Significant Confidence Change Rate (SCCR).

- **Thresholds:** $\epsilon = 0.01$ for minor changes, $\delta = 0.2$ for significant changes.
- **Scaling factor:** $s = 5$ to amplify ACCM and SCCR values.
- **Weights:** $w_1 = 0.4$, $w_2 = 0.4$, $w_3 = 0.2$.

The combined CRS is computed as:

$$\text{CRS} = w_1 \cdot (1 - \text{CCR}) + w_2 \cdot (1 - s \cdot \text{ACCM}) + w_3 \cdot (1 - s \cdot \text{SCCR}).$$

**Theoretical range:** $[-2.4, 1]$, where 1 indicates perfect confidence robustness.

### C.2  CONFIDENCE SENSITIVITY SCORE (CSS)

CSS quantifies how sensitively the confidence score responds to reasoning errors. For each error type $t$:

$$\Delta p_t = \overline{p}_{\text{correct}} - \overline{p}_t, \quad \text{CSS} = \frac{1}{|\mathcal{T}|} \sum_{t \in \mathcal{T}} \Delta p_t,$$

where $\overline{p}_t$ is the average predicted probability for error type $t$, and $\mathcal{T}$ is the set of all error types.

**Range:** $\text{CSS} \in [-1, 1]$. Larger CSS indicates stronger sensitivity to errors.

### C.3  CONFIDENCE CALIBRATION SCORE (CCS)

CCS evaluates calibration quality by combining global ECE and class-wise calibration gap $\Delta\text{ECE}_{\text{cls}}$:

$$\text{CCS} = 0.5 \cdot (1 - s \cdot \text{ECE}) + 0.5 \cdot (1 - \Delta\text{ECE}_{\text{cls}}),$$

with scaling factor $s = 5$.

**Range:** $[-2, 1]$, where higher CCS reflects better calibration.

### C.4  SUMMARY OF PARAMETER CHOICES

All three metrics use parameters chosen based on theoretical and practical considerations:

- CRS thresholds $\epsilon$ and $\delta$ distinguish minor versus significant confidence changes.
- Scaling factor $s$ ensures smaller sub-metrics (ACCM, SCCR, ECE) contribute comparably.

- Weights $(w_1, w_2, w_3)$ balance sub-metric contributions in CRS; CSS and CCS use uniform averaging.

These settings suppress noise, balance contributions across sub-components, and maintain fine-grained interpretability of step-level reasoning.

# D    PROMPT FOR ADVERSARIAL PERTURBATIONS GENERATION AND PROCESS JUDGING

The prompt used to generate reasoning steps with syntactic transformation perturbations is shown in Table 7. The prompt used to generate reasoning steps with synonym substitution perturbations is shown in Table 8. The prompt used for the multimodal process judging task is shown in Table 9.

# E    CONFIDENCE ROBUSTNESS ACROSS PERTURBATION TYPES

As shown in Figure 4, among all types of adversarial perturbations, MPJs exhibit the lowest confidence robustness scores (CRS) under syntactic transformations. This suggests that MPJs are least robust when facing syntactic transformations but semantically equivalent inputs. In contrast, they demonstrate stronger confidence robustness under synonym substitution and image perturbation. These results indicate that MPJs face considerable challenges in maintaining confidence robustness under syntactic transformations, while other types of adversarial perturbations also present non-negligible effects. Designing targeted strategies to enhance the confidence robustness of MPJs is crucial for obtaining reliable confidence estimates.

# F    CONFIDENCE METRICS ACROSS DIFFICULTY LEVELS, SUBJECTS, AND MODALITIES

## F.1    CONFIDENCE METRIC ANALYSIS ACROSS DIFFERENT DIFFICULTY LEVELS.

The scores of the three core confidence metrics at different difficulty levels are shown in Figure 5. Most MPJs exhibit the highest CSS at the Middle School (Mid) level, with noticeable declines at High School (High) and Competition (Com) levels, though the trend is not strictly monotonic. In contrast, CCS shows a clear and consistent downward trend as difficulty increases, indicating that MPJs become increasingly miscalibrated, assigning overly high confidence to incorrect answers or low confidence to correct ones on harder problems. CRS, however, remains relatively stable across all difficulty levels for most MPJs, suggesting that confidence robustness to adversarial perturbations is not significantly affected by task complexity. These results reveal that while MPJs' sensitivity and calibration degrade under more complex reasoning, their robustness remains largely unaffected, highlighting distinct challenges in improving confidence reliability across different dimensions.

## F.2    CONFIDENCE METRIC ANALYSIS ACROSS DIFFERENT INPUT MODALITIES.

The scores of the three core confidence metrics across different input modalities are shown in Figure 6. CSS shows clear modality dependence: most MPJs achieve higher scores in the Multi-image (Multi) setting than in Single-image (Single) or Pure-text (Pure), indicating that richer visual context enhances sensitivity to prediction correctness. In contrast, CCS remains largely consistent across modalities for most MPJs, suggesting limited influence of input type on calibration. Similarly, CRS scores are highly stable across modalities, indicating that robustness to perturbations is generally unaffected. Overall, input modality notably influences sensitivity, while calibration and robustness remain largely modality-invariant.

## F.3    CONFIDENCE METRIC ANALYSIS ACROSS DIFFERENT SUBJECT DOMAINS.

The scores of the three core confidence metrics across different subject domains are shown in Figure 7. The performance of different MPJs on the Confidence Sensitivity Score (CSS) varies across

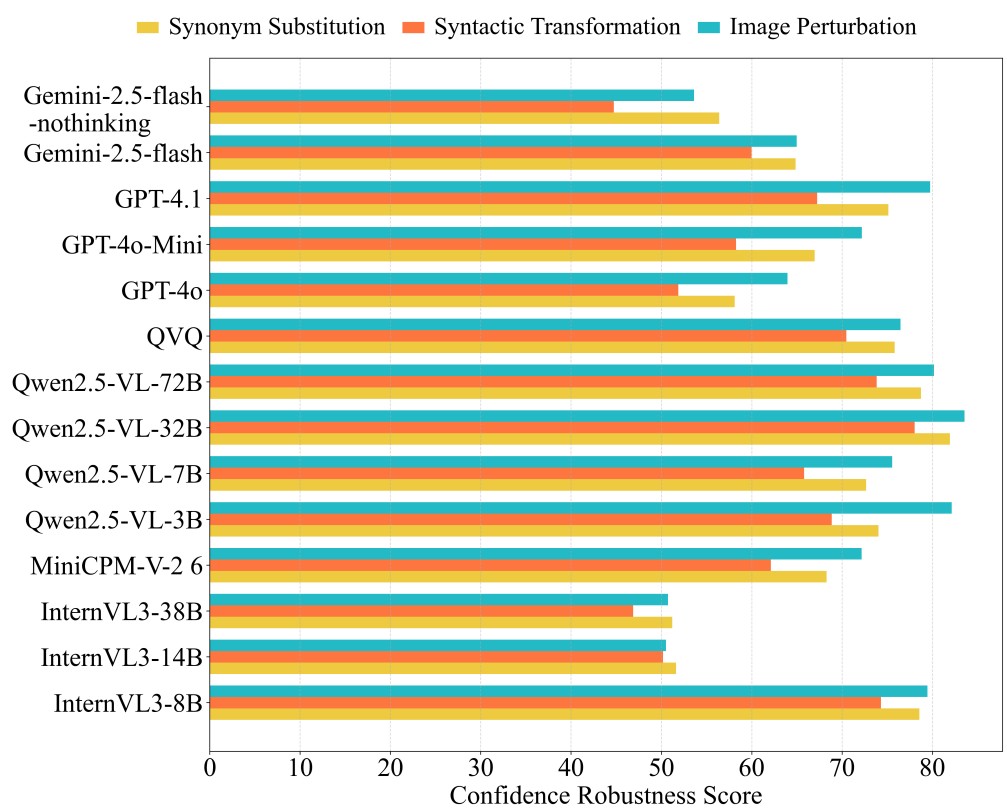

Figure 4: Confidence Robustness Score (CRS) under Different Perturbations

subjects, but no consistent subject-specific trend is observed. This suggests that CSS is more dependent on model-specific characteristics rather than being driven by subject domain, implying that each MPJ may possess unique strengths and weaknesses when handling different types of knowledge structures or symbolic reasoning. Most MPJs achieve higher Confidence Calibration Scores (CCS) in the Biology domain, indicating better alignment between confidence and prediction correctness in that subject. In contrast, Confidence Robustness Scores (CRS) remain highly consistent across all subjects and MPJs, with radar plots forming near-square shapes, suggesting that subject domain has minimal impact on robustness. Overall, MPJs maintain consistent robustness against perturbations across tasks from different subject domains.

## G HIGH CLASSIFICATION PERFORMANCE DOES NOT ENSURE CONFIDENCE RELIABILITY.

As shown in Table 10, strong classification performance of MPJs does not necessarily imply high confidence reliability. For instance, GPT-4o achieves a solid Macro F1 score of 78.12, indicating strong classification ability, yet its confidence sensitivity (CSS = 30.71) and calibration (CCS = 62.00) remain moderate. Similarly, while Gemini-2.5-flash attains the highest Macro F1 (81.74), its CCS (48.62) and robustness (CRS = 63.08) are not the best, revealing a mismatch between classification accuracy and confidence reliability. In contrast, GPT-4.1 demonstrates a more balanced profile, combining a high Macro F1 (80.87) with strong robustness (CRS = 73.62) and sensitivity (CSS = 38.51), though its CCS is relatively lower (37.65).

**Confidence Sensitivity Score (CSS).** We propose Confidence Sensitivity Score (CSS), a novel metric that quantifies how sensitively confidence scores respond to reasoning errors.

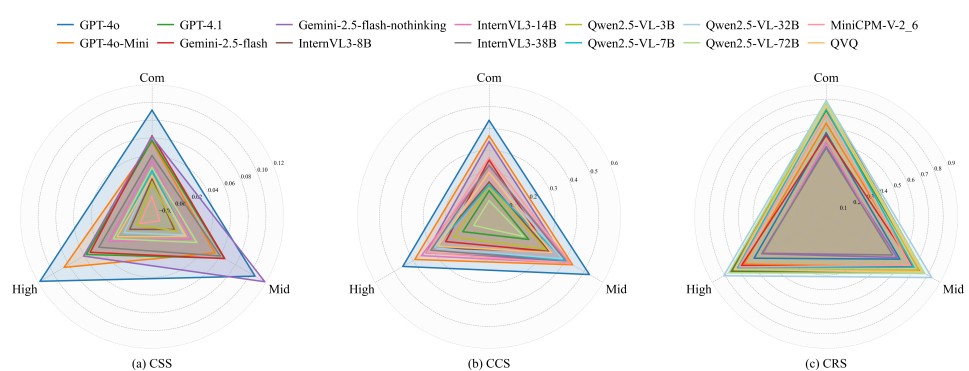

Figure 5: Confidence metric performance of MPJs across different difficulty levels.

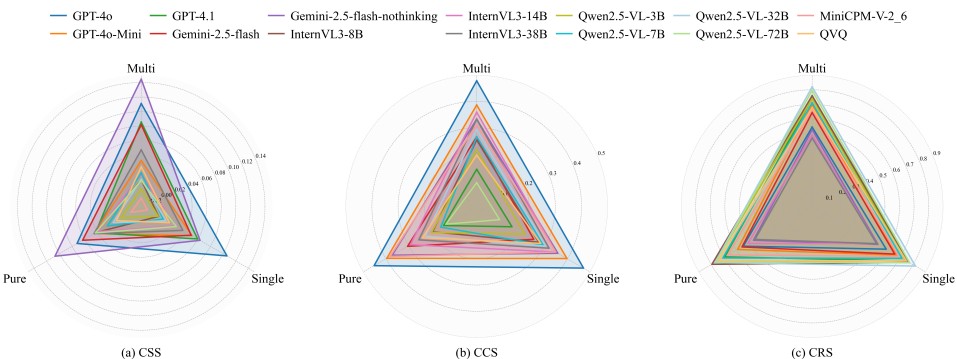

Figure 6: Confidence metric performance of MPJs across different input modalities.

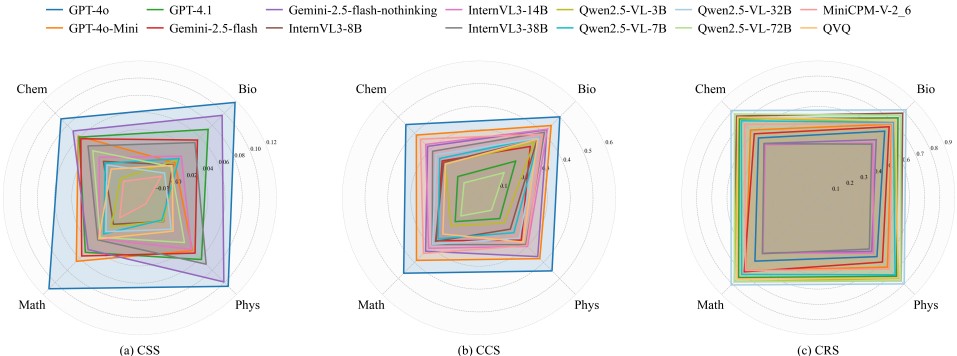

Figure 7: Confidence metric performance of MPJs across different subject domains.

| Statistic | Number |
|---|---|
| Total Samples | 1200 |
| Synonym Replacement / Sentence structure / Image Perturbations | 400 / 400 / 400 |
| Middle School / High School / Competition | 400 / 400 / 400 |
| Math / Physics / Chemistry / Biology | 400 / 400 / 400 |
| Single Image / Multi Images / Pure Text | 823 / 162 /215 |

Table 3: Statistics of ConfProBench

| Model | CCR↓ | ACCM↓ | SCCR↓ |
|---|---|---|---|
| **Open-source MLLMs** | | | |
| InternVL3-8B | 21.47 | 6.56 | 0.89 |
| InternVL3-14B | 61.57 | 9.35 | 5.89 |
| InternVL3-38B | 61.71 | 9.26 | 6.87 |
| MiniCPM-V-2_6 | 22.49 | 10.68 | 1.60 |
| Qwen2.5-VL-3B | **15.78** | 8.65 | 1.69 |
| Qwen2.5-VL-7B | 24.97 | 8.31 | 2.19 |
| Qwen2.5-VL-32B | 15.83 | **6.29** | **0.04** |
| Qwen2.5-VL-72B | 21.81 | 6.63 | 0.58 |
| QVQ | 28.68 | 6.74 | 0.86 |
| **Proprietary MLLMs** | | | |
| GPT-4o | **21.82** | 13.46 | 6.98 |
| GPT-4o-Mini | 27.68 | 9.55 | 4.24 |
| GPT-4.1 | 34.96 | **5.47** | **1.46** |
| Gemini-2.5-flash | 38.56 | 8.17 | 5.15 |
| Gemini-2.5-flash-nothinking | 40.31 | 11.77 | 9.12 |

Table 4: The results of the sub-metrics that constitute the Confidence Robustness Score (CRS). The best performance for each metric is shown in bold, while the second-best is underlined.

# H EVALUATING CONFPROBENCH'S ROBUSTNESS TO DIFFERENT PERTURBATION GENERATORS

To evaluate whether the perturbation construction process is fundamentally constrained by the capability of a specific MLLM, we conducted a **cross-model perturbation robustness study**. This experiment assesses whether perturbations generated by different MLLMs lead to consistent evaluation outcomes on ConfProBench.

## EXPERIMENTAL SETUP

To directly test whether perturbation generation depends on the underlying model, we performed the following controlled experiment:

- We randomly sampled **300** items from the full benchmark.
- For each item, we generated perturbations using two different MLLMs:
    1. GPT-4o
    2. Gemini-2.5-Pro
- For each model, we produced perturbations covering all three perturbation types used in ConfProBench:
    - image-based perturbations (100 samples),
    - syntactic rewriting (100 samples),
    - synonym substitution (100 samples).

| Model | $\Delta p_{\text{NSPE}}$ ↑ | $\Delta p_{\text{RE}}$ ↑ | $\Delta p_{\text{NCE}}$ ↑ | $\Delta p_{\text{SCE}}$ ↑ | $\Delta p_{\text{KE}}$ ↑ | $\Delta p_{\text{VIE}}$ ↑ | $\Delta p_{\text{QUE}}$ ↑ |
|---|---|---|---|---|---|---|---|
| **Open-source MLLMs** | | | | | | | |
| InternVL3-8B | 8.80 | 13.48 | 4.40 | 7.59 | 18.86 | 6.36 | 21.36 |
| InternVL3-14B | 10.24 | **28.28** | 28.35 | 19.58 | 22.79 | 15.26 | **23.85** |
| InternVL3-38B | **60.46** | 35.14 | **31.79** | **24.81** | **32.84** | **17.94** | 11.38 |
| MiniCPM-V-2_6 | 19.95 | 10.01 | 13.23 | 15.46 | 2.03 | 7.12 | -21.62 |
| Qwen2.5-VL-3B | 12.40 | 1.67 | 0.02 | 2.12 | 8.35 | 1.73 | -4.22 |
| Qwen2.5-VL-7B | 18.96 | 10.66 | 5.32 | 5.78 | 15.48 | 8.59 | 7.87 |
| Qwen2.5-VL-32B | 11.23 | 25.60 | 18.59 | 17.87 | 16.79 | 7.76 | 13.64 |
| Qwen2.5-VL-72B | 17.18 | 28.24 | 21.43 | 20.16 | 23.90 | 9.10 | 19.47 |
| QVQ | 24.98 | 12.91 | 6.77 | 14.28 | 14.01 | 6.81 | 8.49 |
| **Proprietary MLLMs** | | | | | | | |
| GPT-4o | **48.34** | 35.12 | 34.73 | 32.53 | 28.67 | 21.53 | 14.01 |
| GPT-4o-Mini | 7.22 | 18.28 | 13.67 | 23.08 | 11.83 | 7.35 | 9.81 |
| GPT-4.1 | 2.38 | 51.77 | **56.23** | 45.68 | 45.15 | 40.83 | 27.49 |
| Gemini-2.5-flash | 27.60 | **54.03** | 54.08 | **53.99** | **53.89** | **49.48** | 44.94 |
| Gemini-2.5-flash-nothinking | 23.17 | 49.34 | 41.57 | 46.61 | 41.85 | 40.57 | **51.77** |

Table 5: The results of the sub-metrics that constitute the Confidence Sensitivity Score (CSS). The best performance for each metric is shown in bold, while the second-best is underlined. NCE denotes Numerical Calculation Error, RE denotes Reasoning Error, SCE denotes Symbolic Calculation Error, KE denotes Knowledge Error, VIE denotes Visual Interpretation Error, QUE denotes Question Understanding Error, and NSPE denotes No Solution Provided Error.

| Model | ECE(C.)↓ | ECE(I.)↓ | $\Delta$ECE↓ | ECE↓ |
|---|---|---|---|---|
| **Open-source MLLMs** | | | | |
| InternVL3-8B | 8.86 | 90.18 | 81.32 | 13.35 |
| InternVL3-14B | 10.71 | 85.05 | 74.34 | **6.43** |
| InternVL3-38B | 8.80 | 84.82 | 76.03 | 7.00 |
| MiniCPM-V-2_6 | 16.51 | **84.61** | 68.09 | 45.16 |
| Qwen2.5-VL-3B | 9.24 | 90.50 | 81.26 | 22.66 |
| Qwen2.5-VL-7B | 9.24 | 88.85 | 79.62 | 17.76 |
| Qwen2.5-VL-32B | 9.41 | 88.88 | 79.47 | 7.47 |
| Qwen2.5-VL-72B | **4.37** | 92.16 | 87.78 | 12.32 |
| QVQ | 8.25 | 89.72 | 81.47 | 11.43 |
| **Proprietary MLLMs** | | | | |
| GPT-4o | 10.54 | **76.93** | 66.39 | **1.92** |
| GPT-4o-Mini | 10.32 | 83.33 | 73.01 | 6.31 |
| GPT-4.1 | **3.00** | 89.81 | 86.81 | 7.58 |
| Gemini-2.5-flash | 6.43 | 87.56 | 81.13 | 4.32 |
| Gemini-2.5-flash-nothinking | 9.06 | 82.02 | 72.97 | 4.79 |

Table 6: The results of the sub-metrics that constitute the Confidence Calibration Score (CCS). C. indicates the correct class, and I. indicates the incorrect class. The best performance for each metric is shown in bold, while the second-best is underlined.

- Both perturbed datasets were evaluated using representative MLLM-based process judges (MPJs), and we computed the three proposed confidence metrics: CRS, CSS, and CCS.

This yielded two parallel perturbed datasets—one constructed using GPT-4o and one using Gemini-2.5-Pro—allowing us to directly measure the consistency of metric outcomes across perturbation sources.

You are a sentence structure rewriting assistant. Your task is to rewrite a given sentence while altering its structure, ensuring that the original meaning is preserved. For each sentence, you must generate five distinct rewritten versions, each applying only one syntactic transformation. The goal is to create varied sentence structures while maintaining semantic accuracy and natural grammar.

**Syntactic Transformations (Choose One per Rewrite):**

Voice Change (Active ↔ Passive)

2. Adverbial Position Adjustment

3. Clause Order or Structure Change

4. Phrase Structure Simplification or Expansion

5. Inversion or Emphatic Structure

6. Conditional / Purpose / Result Structure Transformation

**Key Constraints:**

- Preserve all steps in multi-step logical reasoning chains.

- Do not omit any mathematical derivations, steps, or intermediate expressions.

- Do not change numbers or mathematical expressions, including LaTeX formulas.

- Preserve meaning, grammar, and naturalness.

- Try to keep the length of the rewritten sentence close to the original (within 2–3 words difference). Avoid significant shortening or lengthening unless necessary for syntactic transformation.

- Only one syntactic transformation type per rewritten sentence.

**Output Format:**

```
{
    "Original Sentence": "The original sentence",
    "Rewritten Sentences": [
        "rewritten sentence 1",
        "rewritten sentence 2",
        "rewritten sentence 3",
        "rewritten sentence 4",
        "rewritten sentence 5"
    ]
}
# Student's solution:  step-by-step student's solution
```

Table 7: Prompt for generating reasoning steps with syntactic transformation perturbations.

**Task Description:** You are a synonym substitution assistant. Given an input sentence, your task is to generate five distinct rewrites. In each version, you must replace at least one non-technical term with an appropriate synonym, and should replace as many non-technical terms as possible. Use different combinations of synonyms while keeping the original sentence structure and meaning intact. All outputs must be grammatically correct and sound natural.

**Definition:** Technical terms refer to specialized vocabulary that is specific to a particular field or discipline and should remain unchanged. These include, but are not limited to: mathematical symbols, scientific terminology, programming syntax, technical jargon, and domain-specific abbreviations.

**Key Constraints:**
- Do not modify any structural elements.
- Do not alter any numbers, numerical values, or mathematical expressions, including both plain numbers and LaTeX formulas.
- Do not change list symbols, bullet points, or any other sequence markers.
- Replace only the natural language content—do not alter formatting, technical terms, or domain-specific vocabulary.
- Ensure all rewritten sentences are grammatically correct, natural, and maintain the original meaning.
- Each rewritten version must replace at least one non-technical word, and should replace as many non-technical words as reasonably possible.

**Output Format:**
Provide your output in the following JSON structure:
{
    "Original Sentence": "The original sentence",
    "Synonym Substitutions": [
        "Synonym Substitution 1",
        "Synonym Substitution 2",
        "Synonym Substitution 3",
        "Synonym Substitution 4",
        "Synonym Substitution 5"
    ]
}

```
# Student's solution:  step-by-step student's solution
```

Table 8: Prompt for generating reasoning steps with synonym substitution perturbations.

You are a teacher skilled in evaluating the intermediate steps of a student's solution to a given problem. You are given a scientific problem, its correct final answer, and a student's step-by-step solution.

Your task is as follows:

1. Carefully solve the problem yourself, using the correct final answer as a hint to guide you to a consistent, correct reasoning path.

2. Then, evaluate the correctness of each step in the student's solution.

**For each step, output:**

- The full original step (as a string)
- A correctness label:
  - 1: if the model believes the step is correct (i.e., if P(correct) $\geq$ 0.5)
  - 0: otherwise
- A probability estimate P(correct) $\in$ (0, 1), representing the model's assessment of the likelihood that the step is correct (correctness label = 1)
- If the step is incorrect (correctness label = 0), also provide:
  - An error category (from the list below):
    - Numerical Calculation Error
    - Symbolic Calculation Error
    - Visual Interpretation Error
    - Reasoning Error
    - Knowledge Error
    - Question Understanding Error
    - No solution provided

**Output Format:**

Wrap your output in this Python list format (and nothing else), enclosed by <evaluation> and </evaluation> tags:

```
<evaluation>
[
["Step 1:  ...", correctness_label, P_correct, "Error type if
incorrect"],
...
]
</evaluation>
```

**Requirements:**

- You must return one and only one evaluation entry per step in the student's solution.
- The number of output entries must exactly match the number of steps (e.g., if the student has 15 steps, your output list must contain 15 entries).
- Do not skip, merge, or summarize steps.
- If the step is correct, use an empty string for the error type: `""`.
- Keep each step as a single complete unit, even if it contains multiple sentences.
- Please evaluate each step one by one. Every step must be assessed and scored individually, even if it is very short. Do not merge, omit, or skip any steps.
- Focus exclusively on the scientific, logical, or mathematical correctness of the solution. Ignore differences in formatting, expression style, specific wording, or presentation order, as long as the reasoning and results are valid.

```
# The given problem:  {problem}
# The Correct Final Answer:  {final answer}
# Student's solution:  step-by-step student's solution
```

Table 9: Prompt for multimodal process judging.

| Model | CRS↑ | CSS↑ | CCS↑ | Avg.↑ | Macro F1↑ |
|---|---|---|---|---|---|
| **Open-source MLLMs** | | | | | |
| InternVL3-8B | 77.41 | 11.55 | 25.97 | 38.31 | 59.21 |
| InternVL3-14B | 50.78 | 21.19 | **46.75** | 39.57 | 70.17 |
| InternVL3-38B | 49.92 | **30.62** | 44.49 | 41.68 | **73.66** |
| MiniCPM-V-2 6 | 68.05 | 6.60 | -47.95 | 8.90 | 38.31 |
| Qwen2.5-VL-3B | 74.71 | 3.15 | 2.73 | 26.86 | 50.90 |
| Qwen2.5-VL-7B | 71.19 | 10.38 | 15.80 | 32.46 | 56.88 |
| Qwen2.5-VL-32B | **81.06** | 15.93 | 41.60 | **46.20** | 67.13 |
| Qwen2.5-VL-72B | 77.45 | 19.93 | 25.30 | 40.89 | 68.33 |
| QVQ | 74.17 | 12.60 | 30.69 | 39.15 | 57.29 |
| **Proprietary MLLMs** | | | | | |
| GPT-4o | 57.37 | 30.71 | **62.00** | 50.03 | 78.12 |
| GPT-4o-Mini | 65.58 | 13.03 | 47.73 | 42.11 | 66.08 |
| GPT-4.1 | **73.62** | 38.51 | 37.65 | 49.93 | 80.87 |
| Gemini-2.5-flash | 63.08 | **48.29** | 48.62 | **53.33** | **81.74** |
| Gemini-2.5-flash-nothinking | 51.20 | 42.13 | 51.55 | 48.29 | 79.02 |

Table 10: Performance comparison across different MLLM-based Process Judges on ConfProBench. The best performance for each metric is shown in bold, while the second-best is underlined.

### RESULTS: HIGH CROSS-MODEL CONSISTENCY

For each MPJ, we compared the metric results obtained under GPT-4o-generated perturbations versus Gemini-generated perturbations. Across all MPJs, we computed Pearson correlations between the two perturbation sources. The results show extremely high agreement:

| Metric | Correlation Across Perturbation Models |
|---|---|
| CRS | **0.992** |
| CSS | **0.979** |
| CCS | **0.985** |

Table 11: Consistency of benchmark metrics between perturbations generated by GPT-4o and Gemini-2.5-Pro.

### CONCLUSION

These results demonstrate that the benchmark outcomes are **highly consistent** across perturbation models. In particular:

- The evaluation metrics remain stable regardless of whether perturbations are generated by GPT-4o or Gemini-2.5-Pro.
- The benchmark does *not* overfit or depend on the perturbation style of a specific model.
- Updating the perturbation generator (e.g., when a stronger model becomes available) is **not necessary**, as it does not alter the evaluation conclusions.

This cross-model perturbation study confirms that ConfProBench is **model-agnostic**, scalable, and robust to changes in upstream perturbation generators.

## I   EVALUATING METRIC STABILITY ACROSS DIFFERENT PERTURBATION MODELS

To systematically examine potential bias from perturbation models, we conducted a cross-perturbation experiment using different MLLMs to generate lexical and syntactic perturbations. In

Table 12: CRS / CSS / CCS scores of representative MPJs under perturbations generated by GPT-4o and Gemini-2.5-Pro.

| Perturbation Model | MPJ | CRS | CSS | CCS |
|---|---|---|---|---|
| GPT-4o | GPT-4o | 0.579 | 0.2764 | 0.5451 |
| Gemini-2.5-Pro | GPT-4o | 0.5727 | 0.2919 | 0.5509 |
| GPT-4o | Gemini-2.5-Pro | 0.7764 | 0.5477 | 0.4085 |
| Gemini-2.5-Pro | Gemini-2.5-Pro | 0.7713 | 0.6061 | 0.4442 |
| GPT-4o | GPT-4.1 | 0.7069 | 0.43 | 0.3846 |
| Gemini-2.5-Pro | GPT-4.1 | 0.7125 | 0.44 | 0.3822 |
| GPT-4o | GPT-5 | 0.6049 | 0.5122 | 0.5666 |
| Gemini-2.5-Pro | GPT-5 | 0.6266 | 0.5012 | 0.5731 |
| GPT-4o | Qwen2.5-VL-32B | 0.8043 | 0.204 | 0.4035 |
| Gemini-2.5-Pro | Qwen2.5-VL-32B | 0.7892 | 0.2263 | 0.4261 |
| GPT-4o | Qwen2.5-VL-72B | 0.781 | 0.134 | 0.2965 |
| Gemini-2.5-Pro | Qwen2.5-VL-72B | 0.7868 | 0.2192 | 0.2744 |

this experiment, we used two heterogeneous MLLMs (GPT-4o and Gemini-2.5-Pro) to generate perturbed datasets, and then evaluated the same set of representative process judges (MPJs) on both datasets to measure stability indicators (CRS, CSS, CCS). This setup allows us to assess: (i) whether GPT-4o perturbations systematically favor OpenAI models, and (ii) whether the evaluation results remain stable across different perturbation sources.

**Experimental Setup.** For each target MPJ, we constructed two perturbed datasets:

- **GPT-4o perturbations**: fully generated by GPT-4o;
- **Gemini-2.5-Pro perturbations**: fully generated by Gemini-2.5-Pro.

We then evaluated the following representative MPJs on both datasets:

GPT-4o, Gemini-2.5-Pro, GPT-4.1, GPT-5, Qwen2.5-VL-32B, Qwen2.5-VL-72B.

The corresponding CRS, CSS, and CCS scores are shown in Table 12.

**Observations (based on Table 12).**

- **GPT-4o perturbations do not systematically favor OpenAI MPJs.** The scores of OpenAI models (GPT-4o, GPT-4.1, GPT-5) under different perturbation sources vary only slightly (typically $< 0.03$), without a consistent upward or downward trend.
- **Gemini MPJs are stable across perturbation sources.** The CRS/CSS/CCS of Gemini-2.5-Pro show minimal differences between the two perturbation sources, indicating that perturbation source has negligible impact on performance.
- **Qwen models also maintain consistent trends.** Both Qwen models show similar performance regardless of perturbation generator, further confirming the robustness of the evaluation.
- **Relative ranking of MPJs is preserved across perturbations.** The relative ordering of all MPJs remains nearly unchanged, demonstrating strong robustness of our evaluation metrics.

**Conclusion.** The cross-perturbation experiment confirms that: Using GPT-4o to generate lexical and syntactic perturbations does not introduce systematic bias favoring OpenAI models. All representative MPJs show stable and consistent CRS, CSS, and CCS scores across different perturbation sources.

