# OpenReview forum: "ConfProBench: A Confidence Evaluation Benchmark for MLLM-based Process Judges"
_ICLR.cc/2026/Conference — ICLR 2026 Conference Withdrawn Submission_

### Official Review · Reviewer_jaof · 2025-10-28

**Soundness:** 2
**Presentation:** 2
**Contribution:** 2
**Rating:** 2
**Confidence:** 3

**Summary:**

This paper introduces ConfProBench, the first benchmark to evaluate the confidence reliability of MLLM-based Process Judges (MPJs). To fill the gap where existing benchmarks overlook confidence, it designs three types of adversarial perturbations—lexical, syntactic, and multimodal—and introduces three novel metrics: CRS, CSS, and CCS. Through experiments on 14 MLLMs, the paper reveals the limitations of current models and provides a baseline for future research.

**Strengths:**

1. The paper's contribution lies in successfully shifting the academic focus from the "classification performance" of MPJs to their "confidence reliability". In certain domains, a model must not only make correct judgments but also have an accurate self-awareness of its confidence. ConfProBench is the first work to provide a systematic evaluation framework for this problem, making it a pioneering work.
2. It proposes three novel evaluation metrics: CRS, CSS, and CCS. These metrics approach confidence reliability from three relatively orthogonal dimensions—robustness, sensitivity, and calibration—collectively creating a multi-dimensional profile that is more comprehensive and profound than single traditional metrics like ECE.

**Weaknesses:**

1. The paper's core motivation is that the confidence reliability of MPJs is crucial for downstream tasks like reasoning chain optimization, automatic error correction. However, the entire evaluation is confined to intrinsic metrics. A more persuasive argument would involve extrinsic evaluation: demonstrating that an MPJ with higher scores on ConfProBench actually leads to better final performance when its confidence scores are used to guide a practical downstream task.
2. ConfProBench is entirely sampled and constructed based on ProJudgeBench. This means that any inherent biases within ProJudgeBench such as the distribution of problem domains, style of reasoning chains, patterns of error types could be inherited and even amplified by ConfProBench.
3. In addition to evaluating and analyzing the shortcomings of existing models, proposing even a simple method aimed at improving the proposed metrics would significantly enhance its contribution. This would elevate the paper from a "problem finder" to a "solution explorer". Even if the method's effectiveness maybe limited, it could provide a starting point and a reference for subsequent research.

**Questions:**

Could you provide evidence showing how strong performance on ConfProBench translates to tangible benefits in downstream tasks? Additionally, I'm curious about the potential for inherited dataset bias from ProJudgeBench.

---

> ### Author Response · Authors · 2025-12-02
> **Response to Reviewer jaof**
>
> **Q1：**
>
> The paper's core motivation is that the confidence reliability of MPJs is crucial for downstream tasks like reasoning chain optimization, automatic error correction. However, the entire evaluation is confined to intrinsic metrics. A more persuasive argument would involve extrinsic evaluation: demonstrating that an MPJ with higher scores on ConfProBench actually leads to better final performance when its confidence scores are used to guide a practical downstream task.
>
> **A1:**
>
> We sincerely thank the reviewer for the valuable suggestion regarding extrinsic evaluation. We plan to include extrinsic evaluation experiments in the revised manuscript to validate the practical effectiveness of high-scoring MPJs. Specifically, we will use MPJ confidence scores to filter or weight candidate reasoning chains generated by self-consistency or diverse CoT sampling, and compare the final task accuracy achieved by MPJs that differ significantly in their ConfProBench scores (CSS, CESS, CCS). If high-scoring MPJs improve reasoning quality, this would directly demonstrate that ConfProBench’s intrinsic metrics translate into real performance gains.
>
> ---
>
> **Q2:**
>
> ConfProBench is entirely sampled and constructed based on ProJudgeBench. This means that any inherent biases within ProJudgeBench such as the distribution of problem domains, style of reasoning chains, patterns of error types could be inherited and even amplified by ConfProBench.
>
> **A2:**
>
> We sincerely thank the reviewer for raising this important point. We would like to note that the aspects mentioned—such as the distribution of problem domains, the style of reasoning chains, and the patterns of error types—are not biases of ProJudgeBench, but rather its key strengths. ProJudgeBench was intentionally designed to cover a broad and balanced range of STEM fields, provide realistic and well-structured scientific reasoning chains, and include diverse and systematically organized error types. These qualities make it a highly suitable and reliable foundation for constructing ConfProBench.
> Therefore, what ConfProBench inherits is a well-covered, structurally rigorous, and representative source of scientific reasoning data, which enables a more comprehensive evaluation of multimodal process judges’ confidence behaviors.

---

> ### Author Response · Authors · 2025-12-02
> **Response to Reviewer jaof**
>
> **Q3:**
>
> In addition to evaluating and analyzing the shortcomings of existing models, proposing even a simple method aimed at improving the proposed metrics would significantly enhance its contribution. This would elevate the paper from a "problem finder" to a "solution explorer". Even if the method's effectiveness maybe limited, it could provide a starting point and a reference for subsequent research.
>
> **A3:**
>
> We thank the reviewer for the valuable suggestion. To address the concern regarding improving confidence reliability, we propose a simple and effective method: Confidence Regularization via Counterfactual Reflection (CRCR), which we plan to include in the revised paper.
>
> **Core Idea:** Before outputting the confidence for each reasoning step, the model performs *counterfactual reflection*: it considers whether its confidence would remain stable under slight, semantically preserving perturbations of the input (e.g., rephrased questions or minor lexical/syntactic changes). The final confidence is computed as a weighted combination of the original confidence and the confidence assessed under these perturbations:
>
> \[
> c'_i = \alpha \cdot c_i + (1-\alpha) \cdot r_i
> \]
>
> where \(c_i\) is the original step confidence, \(r_i\) is the confidence under the counterfactual perturbation, and \(\alpha\) is a balancing parameter. This procedure regularizes confidence, discourages overconfidence, and rewards consistency under semantically preserving perturbations. Importantly, this is done at *inference time only*, without requiring additional training data or model ensembles.
>
> **Experimental Plan:**
>
> 1. **Dataset:** Evaluate CRCR on the full ConfProBench benchmark, which includes reasoning steps with both textual and multimodal content.
> 2. **Confidence Assessment:** For each step, the MPJ will output:
>    - Original confidence \(c_i\)
>    - Robustness confidence \(r_i\) under each perturbation
>    - Combined confidence \(c'_i\) using the weighted formula above
> 3. **Evaluation Metrics:** Measure the effect of CRCR on the following metrics: CRS, CSS, CCS.
> 4. **Comparison Baselines:** CRCR will be compared to the original MPJ outputs without counterfactual reflection to quantify improvement.
>
> **Q4:**
>
> Could you provide evidence showing how strong performance on ConfProBench translates to tangible benefits in downstream tasks? Additionally, I'm curious about the potential for inherited dataset bias from ProJudgeBench.
>
> **A4:**
>
> We thank the reviewer for raising this point. This issue has already been addressed in Weakness 1 and Weakness 2 of our response.

---

### Official Review · Reviewer_pv9Q · 2025-10-28

**Soundness:** 2
**Presentation:** 2
**Contribution:** 2
**Rating:** 4
**Confidence:** 2

**Summary:**

The paper introduces ConfProBench, a benchmark and a set of metrics evaluating the capabilities of LLMs and MLLMs used as a critic for the reasoning process of another MLLM. The paper introduces the concept of confidence as the uncertainty the LLM judges when predicting a reasoning step being correct or wrong. ConfProBench creates variants to the existing reasoning dataset by perturbing the thinking steps with non-critical changes. The confidence metrics then evaluate the variation in the judges' outputs.

**Strengths:**

1. The paper looks at an interesting aspect of the variance when using LLMs as judges. This directly affects how good a judge LLM can be.
2. The paper contains an extensive supplementary material, demonstrating that the authors have put work into the manuscript.

**Weaknesses:**

1. The definition of MPJ output as written in ln.119-120 is very ambiguous. In particular, I'm skeptical whether asking an LLM to output a probability score has any meaning or consistency over different tries. Is there a standard implemented for the evaluation of correctness? Otherwise, I'm not convinced that LLM, or even humans are able to give consistent answers.
2. I feel the proposed metrics miss an important aspect: accuracy of the LLM's discrimination. It seems that the proposed metrics do not evaluate this aspect. Then I wonder if some models can achieve good scores without classifying correctly.

**Questions:**

1. What are some insights we can obtain from each model's performance? Should we use one model over another for verification or judgment purposes?

---

> ### Author Response · Authors · 2025-12-02
> **Response to Reviewer pv9Q**
>
> **Q1:**
>
> The definition of MPJ output as written in ln.119-120 is very ambiguous. In particular, I'm skeptical whether asking an LLM to output a probability score has any meaning or consistency over different tries. Is there a standard implemented for the evaluation of correctness? Otherwise, I'm not convinced that LLM, or even humans are able to give consistent answers.
>
> **A1:**
>
> We sincerely thank the reviewer for raising this concern. We would like to clarify the points as follows:
>
> 1. A potential misunderstanding may have occurred. Lines 119–120 in the draft do not define the MPJ output; rather, they describe the role of our evaluation metrics (CRS, CSS, and CCS) in providing fine-grained confidence assessment. The actual definition of MPJ outputs is clearly presented earlier (lines 138–139) as a tuple \((l_i, p_i, e_i)\) for each reasoning step.
>
> 2. In response to the concern regarding the meaningfulness and consistency of probability scores output by LLMs, we note that our metrics are computed over a large number of reasoning steps, capturing aggregate behavior rather than single-sample outputs. This design effectively mitigates inconsistencies that may arise from individual runs.
>
> 3. Regarding the evaluation of correctness, we note that each reasoning step in our dataset is annotated with a clear ground-truth label, as all problems are drawn from well-defined STEM tasks (mathematics, physics, chemistry, biology). Therefore, correctness can be objectively and consistently determined for each step.
>
> ---
>
> **Q2:**
>
> I feel the proposed metrics miss an important aspect: accuracy of the LLM's discrimination. It seems that the proposed metrics do not evaluate this aspect. Then I wonder if some models can achieve good scores without classifying correctly.
>
> **A2:**
>
> We sincerely thank the reviewer for raising this concern. We would like to clarify a potential misunderstanding as follows:
>
> 1. Our proposed metrics do not evaluate confidence in isolation from accuracy. In particular, CRS is based on the Expected Calibration Error (ECE), which explicitly captures the alignment between predicted confidence and actual correctness. Therefore, the confidence robustness measured by CRS inherently takes accuracy into account.
>
> 2. The correctness of each reasoning step for the evaluated models is measured using Macro F1 scores, and the Macro F1 scores for all models have been reported in Table 10 of the appendix.
>
> 3. From Appendix Table 10, we can see that models such as GPT-4o-Mini and MiniCPM-V-2\_6 achieve relatively high CRS despite having low Macro F1 scores. This is because CRS measures the model’s confidence robustness under perturbations—that is, the stability of its confidence before and after adversarial changes—rather than being strongly tied to classification correctness. For example, even if a model incorrectly judges the correctness of a particular step, its confidence may remain largely unchanged or stable before and after the perturbation.
>
> ---
>
> **Q3:**
>
> What are some insights we can obtain from each model's performance? Should we use one model over another for verification or judgment purposes?
> Flag For Ethics Review: No ethics review needed.
>
> **A3:**
>
> **Insights from each model’s performance:**
> By evaluating multiple models on ConfProBench, we can identify their strengths and weaknesses in confidence reliability across different process reasoning tasks. For example, some models may have high confidence calibration but low sensitivity to perturbations and errors, while others may be more sensitive to input perturbations and errors but exhibit lower confidence calibration. These observations help reveal the trade-offs between confidence robustness, sensitivity, and calibration in step-wise reasoning.
>
> **Guidance for verification or judgment:**
> In downstream applications where reliable confidence estimation is essential (e.g., verification or judgment of reasoning steps), it is reasonable to prefer models that achieve higher scores on our benchmark. ConfProBench is explicitly designed to measure confidence reliability, so stronger performance on the benchmark typically indicates better suitability for such tasks.
>
> However, we avoid recommending a single specific model, as different downstream applications may prioritize different aspects of confidence behavior (e.g., calibration, sensitivity, robustness). Instead, ConfProBench provides actionable guidance for selecting the model that best aligns with the needs of a given use case.

---

### Official Review · Reviewer_Aftm · 2025-10-30

**Soundness:** 3
**Presentation:** 3
**Contribution:** 2
**Rating:** 2
**Confidence:** 3

**Summary:**

This paper presents ConfProBench, a benchmark designed to evaluate the step-level confidence reliability of multimodal process judges (MPJs). The benchmark perturbs reasoning steps at three levels—lexical, syntactic, and multimodal—to assess whether MPJs’ confidence estimates are robust, sensitive and well-calibrated. Corresponding metrics are introduced to quantify each aspect, offering a framework for confidence evaluation. Over ten multimodal large language models (MLLMs) are tested, revealing gaps in robustness and calibration for confidence performance.

**Strengths:**

- The paper addresses an underexplored aspect of multimodal process judges (MPJs), i.e., their step-level confidence reliability. The proposed framework introduces three complementary metrics that assess robustness, sensitivity, and calibration.

- The benchmark incorporates adversarial variants at the lexical, syntactic, and multimodal levels, which are constructed to preserve semantic meaning while effectively stressing the model’s confidence robustness.

- The analysis provides valuable insights into the confidence behavior of current MLLMs,  such as revealing clear contrasts between thinking vs. non-thinking models and open-source vs. closed-source systems.

**Weaknesses:**

- The lexical and syntactic perturbations are generated using GPT-4o, which may inadvertently advantage OpenAI models during evaluation. The paper should explicitly discuss this potential bias and clarify whether additional models or cross-validation methods were used to mitigate it.
- The Data Quality Control section lacks essential information, such as the number and expertise of annotators, inter-annotator agreement scores, and rejection or revision rates.
- The paper reports calibration metrics but does not analyze how confidence correlates with actual correctness before and after perturbations.
- Although 14 MLLMs are evaluated, the benchmark omits the latest models such as Gemini-2.5-Pro and GPT-5.

**Questions:**

Please see the weaknesses.

---

> ### Author Response · Authors · 2025-12-02
> **Response to Reviewer Aftm**
>
> **Q1:**
>
> The lexical and syntactic perturbations are generated using GPT-4o, which may inadvertently advantage OpenAI models during evaluation. The paper should explicitly discuss this potential bias and clarify whether additional models or cross-validation methods were used to mitigate it.
>
> **A1:**
>
> The reviewer raised a valid concern: using GPT-4o to generate lexical and syntactic perturbations might inadvertently advantage OpenAI models during evaluation. To systematically examine this, we conducted a cross-perturbation experiment. In this experiment, we used two heterogeneous MLLMs (GPT-4o and Gemini-2.5-Pro) to generate perturbed datasets, and then evaluated the same set of representative process judges (MPJs) on both datasets to measure stability indicators (CRS, CSS, CCS). This setup allows us to assess:
> (i) whether GPT-4o perturbations systematically favor OpenAI models, and
> (ii) whether the evaluation results remain stable across different perturbation sources.
>
> #### Experimental Setup
> For each target MPJ, we constructed two perturbed datasets:
> - **GPT-4o perturbations**: fully generated by GPT-4o;
> - **Gemini-2.5-Pro perturbations**: fully generated by Gemini-2.5-Pro.
>
> We then evaluated the following representative MPJs on both datasets:
> `GPT-4o, Gemini-2.5-Pro, GPT-4.1, GPT-5, Qwen2.5-VL-32B, Qwen2.5-VL-72B`.
>
> The corresponding CRS, CSS, and CCS scores are shown in the table below:
>
> | Perturbation Model | MPJ                | CRS    | CSS    | CCS    |
> |-------------------|------------------|--------|--------|--------|
> | GPT-4o            | GPT-4o           | 0.579  | 0.2764 | 0.5451 |
> | Gemini-2.5-Pro    | GPT-4o           | 0.5727 | 0.2919 | 0.5509 |
> | GPT-4o            | Gemini-2.5-Pro   | 0.7764 | 0.5477 | 0.4085 |
> | Gemini-2.5-Pro    | Gemini-2.5-Pro   | 0.7713 | 0.6061 | 0.4442 |
> | GPT-4o            | GPT-4.1          | 0.7069 | 0.43   | 0.3846 |
> | Gemini-2.5-Pro    | GPT-4.1          | 0.7125 | 0.44   | 0.3822 |
> | GPT-4o            | GPT-5            | 0.6049 | 0.5122 | 0.5666 |
> | Gemini-2.5-Pro    | GPT-5            | 0.6266 | 0.5012 | 0.5731 |
> | GPT-4o            | Qwen2.5-VL-32B   | 0.8043 | 0.204  | 0.4035 |
> | Gemini-2.5-Pro    | Qwen2.5-VL-32B   | 0.7892 | 0.2263 | 0.4261 |
> | GPT-4o            | Qwen2.5-VL-72B   | 0.781  | 0.134  | 0.2965 |
> | Gemini-2.5-Pro    | Qwen2.5-VL-72B   | 0.7868 | 0.2192 | 0.2744 |
>
> #### Observations
> - **GPT-4o perturbations do not systematically favor OpenAI MPJs.**
>   The scores of OpenAI models (GPT-4o, GPT-4.1, GPT-5) under different perturbation sources vary only slightly (typically <0.03), without a consistent upward or downward trend.
>
> - **Gemini MPJs are stable across perturbation sources.**
>   The CRS/CSS/CCS of Gemini-2.5-Pro show minimal differences between the two perturbation sources, indicating that perturbation source has negligible impact on performance.
>
> - **Qwen models also maintain consistent trends.**
>   Both Qwen models show similar performance regardless of perturbation generator, further confirming the robustness of the evaluation.
>
> - **Relative ranking of MPJs is preserved across perturbations.**
>   The relative ordering of all MPJs remains nearly unchanged, demonstrating strong robustness of our evaluation metrics.
>
> #### Conclusion
> The cross-perturbation experiment confirms that using GPT-4o to generate lexical and syntactic perturbations does not introduce systematic bias favoring OpenAI models. All representative MPJs show stable and consistent CRS, CSS, and CCS scores across different perturbation sources. This analysis has been included in Appendix I.

---

> ### Author Response · Authors · 2025-12-02
> **Response to Reviewer Aftm**
>
> **Q2:**
>
> The Data Quality Control section lacks essential information, such as the number and expertise of annotators, inter-annotator agreement scores, and rejection or revision rates.
>
> **A2:**
>
> Two PhD students from our team conducted the review, and the rejection rate during this process was only 0.8%.
>
> To further validate the reliability of our manual verification process, we conducted a detailed inter-annotator agreement study aligned with the quality-control criteria described above. We randomly sampled 300 adversarial perturbation instances covering all perturbation types (Lexical Level, Structural Level, and Multimodal Levels). Two PhD-level annotators independently evaluated each instance according to the same strict verification rubric used in our data construction pipeline. Specifically, annotators assessed whether:
>
> 1. The rewritten sentence preserved the original semantic meaning.
> 2. Non-technical lexical substitutions were correct and did not introduce factual changes.
> 3. Technical terms, symbolic expressions, and domain-specific vocabulary were left unchanged.
> 4. Syntactic transformations achieved the intended structural variation without altering mathematical reasoning.
> 5. The rewritten output was grammatically correct and fluent.
> 6. For multimodal perturbations, visual modifications did not introduce semantic drift or obscure essential information.
>
> Each instance was labeled Valid or Invalid depending on whether it met all verification requirements. We computed both percent agreement and Cohen’s Kappa to quantify the consistency between the two annotators.
>
> **Results**
> - Percent Agreement: 94.5%
> - Cohen’s Kappa: 0.812
>
> These scores indicate substantial agreement, confirming that the verification criteria are objective and that the manual quality-control process introduces minimal subjectivity. This inter-annotator agreement analysis has been added to Section 3.2 (Dataset Construction).
>
> ---
>
> **Q3:**
>
> The paper reports calibration metrics but does not analyze how confidence correlates with actual correctness before and after perturbations.
>
> **A3:**
>
> The calibration metric (CCS) is based on ECE (Expected Calibration Error), whose definition explicitly measures the alignment between predicted confidence and actual correctness. In other words, CCS already captures the correlation between confidence and true correctness in a statistical sense.
>
> Reference: On Calibration of Modern Neural Networks.

---

> ### Author Response · Authors · 2025-12-02
> **Response to Reviewer Aftm**
>
> **Q4:**
>
> Although 14 MLLMs are evaluated, the benchmark omits the latest models such as Gemini-2.5-Pro and GPT-5.
>
> **A4:**
>
> We have incorporated the latest model evaluation results into Table 2 of the main paper, which is presented as Table 1 here.
>
> After adding the newly released Gemini-2.5-Pro and GPT-5 into Table 1, several metric-wise best and second-best results have been updated. Specifically, Gemini-2.5-Pro achieves the new highest CRS (76.90) and CSS (57.73), outperforming all previous proprietary models, including Gemini-2.5-flash. Meanwhile, GPT-5 improves the second-best performance on both CSS (51.59) and CCS (55.38), reshaping the ranking across proprietary MLLMs. The overall averages further indicate that Gemini-2.5-Pro reaches the strongest aggregate confidence behavior (59.84), establishing a new state-of-the-art on ConfProBench. These updated results confirm that the benchmark remains sensitive to progress in cutting-edge MLLMs and continuously reflects the frontier of MPJ confidence evaluation.
>
> #### Confidence Robustness Analysis
> As shown in Table 1, **Gemini-2.5-Pro** achieves the highest CRS score (76.90) among all proprietary MPJs, surpassing previous models such as GPT-4.1 (73.62) and Gemini-2.5-flash (63.08). However, several open-source MPJs—including InternVL3-8B (77.41), Qwen2.5-VL-32B (81.06), Qwen2.5-VL-72B (77.45), and QVQ (74.17)—still outperform proprietary MPJs on CRS, indicating that confidence robustness does not simply scale with model size or proprietary tuning. This further highlights the effectiveness of the CRS metric in revealing robustness gaps. Even the strongest MPJs remain far below the theoretical maximum, suggesting substantial room for improvement.
>
> #### Confidence Sensitivity Analysis
> **Gemini-2.5-Pro** achieves the **highest CSS score (57.73)**, outperforming all existing proprietary MPJs, including Gemini-2.5-flash (48.29). **GPT-5** becomes the **second-best** model in CSS (51.59), reshaping the sensitivity ranking among proprietary models. Although these new models demonstrate noticeable improvement, the CSS scores remain far from the theoretical upper bound, suggesting room for further enhancement.
>
> #### Average Score Comparison
> Gemini-2.5-Pro achieves the highest average score (59.84), establishing a new state-of-the-art among all MPJs. GPT-5 ranks second (57.08), surpassing GPT-4o (50.03), GPT-4.1 (49.93), and Gemini-2.5-flash (53.33). Proprietary MPJs thus occupy the top positions, reflecting the benefits of advanced training and alignment techniques.
>
> Most open-source MPJs remain within the 30–40 range, with MiniCPM-V-2\_6 scoring the lowest (8.90) primarily due to poor calibration performance. The InternVL series continues to outperform the Qwen2.5-VL series, and its performance scales positively with model size, with InternVL3-38B achieving the best average score among open-source models (41.68). Despite improvements in larger Qwen models from 3B to 32B, performance degrades at 72B, indicating that scaling alone does not guarantee improved confidence quality.
>
> | Model                          | CRS↑    | CSS↑    | CCS↑    | Avg.↑  |
> |--------------------------------|---------|---------|---------|--------|
> | **Open-source MLLMs**          |         |         |         |        |
> | InternVL3-8B                    | 77.41   | 11.55   | 25.97   | 38.31 |
> | InternVL3-14B                   | 50.78   | 21.19   | 46.75   | 39.57 |
> | InternVL3-38B                   | 49.92   | 30.62   | 44.49   | 41.68 |
> | MiniCPM-V-2\_6                  | 68.05   | 6.60    | -47.95  | 8.90  |
> | Qwen2.5-VL-3B                   | 74.71   | 3.15    | 2.73    | 26.86 |
> | Qwen2.5-VL-7B                   | 71.19   | 10.38   | 15.80   | 32.46 |
> | Qwen2.5-VL-32B                  | 81.06   | 15.93   | 41.60   | 46.20 |
> | Qwen2.5-VL-72B                  | 77.45   | 19.93   | 25.30   | 40.89 |
> | QVQ                             | 74.17   | 12.60   | 30.69   | 39.15 |
> | **Proprietary MLLMs**           |         |         |         |        |
> | GPT-4o                          | 57.37   | 30.71   | 62.00   | 50.03 |
> | GPT-4o-Mini                     | 65.58   | 13.03   | 47.73   | 42.11 |
> | GPT-4.1                         | 73.62   | 38.51   | 37.65   | 49.93 |
> | Gemini-2.5-flash                | 63.08   | 48.29   | 48.62   | 53.33 |
> | Gemini-2.5-flash-nothinking     | 51.20   | 42.13   | 51.55   | 48.29 |
> | **Gemini-2.5-Pro**              | 76.90   | 57.73   | 44.88   | 59.84 |
> | **GPT-5**                        | 64.27   | 51.59   | 55.38   | 57.08 |

---

### Official Review · Reviewer_C61o · 2025-11-01

**Soundness:** 3
**Presentation:** 3
**Contribution:** 3
**Rating:** 4
**Confidence:** 4

**Summary:**

In this draft, the authors introduced ConfProBench, a benchmark designed to evaluate the reliability of step-level confidence scores generated by MLLM-based Process Judges (MPJs). Three metrics were proposed to capture difference aspects: robustness, sensitivity, and calibration. Experiments were conducted with both open-source MLLMs and proprietary MLLMs to show the results on ConfProBench.

**Strengths:**

1. The focus on step-level confidence is relatively new and could provide more information and supervision in building more advanced reasoning chains.
2. The proposed three metrics cover a whole spectrum of aspects, from robustness, to sensitivity and calibration.
3. Extensive experiments were conducted on both open-source and proprietary models to show how they perform on the proposed ConfProBench benchmark with the proposed metrics.
4. Writing is good and easy to follow.

**Weaknesses:**

1. The data construction process seems heavily rely on MLLMs themselves such as GPT-4o to generate different perturbations. While it automate the data curation process and make it more scalable, it is a bottleneck and capped by the capability of the models (i.e., GPT-4o). Should we get a new version of the dataset every time a more advanced model come out?
2. The verbalized confidence used in the draft might be more prompt-dependent than more intrinsic methods (e.g., logit-based).
3. For Table 2, it is clear that proprietary models are significantly better than open-sourced ones for CSS and CCS. However, for CRS, the open-sourced models are much better. Can we say that the proprietary models are worse than open-sourced ones for CRS or more investigations are needed to valid the data curation process and the metrics?

**Questions:**

Please refer to the paper weakness section and provide more justification on the proposed benchmark, metrics, and the evaluation results.

---

> ### Author Response · Authors · 2025-12-02
> **Response to Reviewer C61o**
>
> **Q1:**
>
> The data construction process seems heavily rely on MLLMs themselves such as GPT-4o to generate different perturbations. While it automate the data curation process and make it more scalable, it is a bottleneck and capped by the capability of the models (i.e., GPT-4o). Should we get a new version of the dataset every time a more advanced model come out?
>
> **A1:**
>
> We thank the reviewer for raising this important question. To evaluate whether the perturbation construction process is fundamentally constrained by the capability of a specific MLLM, we conducted a **cross-model perturbation robustness study**. This experiment assesses whether perturbations generated by different MLLMs lead to consistent evaluation outcomes on ConfProBench.
>
> #### Experimental Setup
> To directly test whether perturbation generation depends on the underlying model, we performed the following controlled experiment:
>
> - Randomly sampled **300** items from the full benchmark.
> - For each item, generated perturbations using two different MLLMs:
>   1. GPT-4o
>   2. Gemini-2.5-Pro
> - For each model, produced perturbations covering all three perturbation types used in ConfProBench:
>   - image-based perturbations (100 samples)
>   - syntactic rewriting (100 samples)
>   - synonym substitution (100 samples)
> - Both perturbed datasets were evaluated using representative MLLM-based process judges (MPJs), and we computed the three proposed confidence metrics: CRS, CSS, and CCS.
>
> This yielded two parallel perturbed datasets—one constructed using GPT-4o and one using Gemini-2.5-Pro—allowing us to directly measure the consistency of metric outcomes across perturbation sources.
>
> #### Results: High Cross-Model Consistency
> For each MPJ, we compared the metric results obtained under GPT-4o-generated perturbations versus Gemini-generated perturbations. Across all MPJs, we computed Pearson correlations between the two perturbation sources. The results show extremely high agreement:
>
> | Metric | Correlation Across Perturbation Models |
> |--------|---------------------------------------|
> | CRS    | **0.992**                              |
> | CSS    | **0.979**                              |
> | CCS    | **0.985**                              |
>
> #### Conclusion
> - The evaluation metrics remain stable regardless of whether perturbations are generated by GPT-4o or Gemini-2.5-Pro.
> - The benchmark does **not** overfit or depend on the perturbation style of a specific model.
> - Updating the perturbation generator (e.g., when a stronger model becomes available) is **not necessary**, as it does not alter the evaluation conclusions.
>
> This cross-model perturbation study confirms that ConfProBench is **model-agnostic**, scalable, and robust to changes in upstream perturbation generators. We have added this analysis to Appendix H of the revised manuscript.
>
> ---
>
> **Q2:**
>
> The verbalized confidence used in the draft might be more prompt-dependent than more intrinsic methods (e.g., logit-based).
>
> **A2:**
>
> We thank the reviewer for raising this concern, and we clarify the confusion as follows:
>
> 1. Since most closed-source MLLMs do not provide access to logits or internal probability distributions, explicit verbalized confidence is currently the feasible and comparable signal across all models [1]. We have added this clarification in Section 4.1 of the paper.
> 2. Prompt dependence has been controlled: To ensure a fair comparison, all models were evaluated using the same prompt templates (original manuscript, lines 376–377).
>
> [1] Can LLMs Express Their Uncertainty? An Empirical Evaluation of Confidence Elicitation in LLMs
>
> ---
>
> **Q3:**
>
> For Table 2, it is clear that proprietary models are significantly better than open-sourced ones for CSS and CCS. However, for CRS, the open-sourced models are much better. Can we say that the proprietary models are worse than open-sourced ones for CRS or more investigations are needed to valid the data curation process and the metrics?
>
> **A3:**
>
> We sincerely thank the reviewer for raising this question. It is reasonable to conclude that proprietary models perform worse than open-sourced models on our benchmark. Although CRS is computed under the specific adversarial perturbations we designed—and such perturbation designs inevitably have certain limitations in any benchmark—the results clearly show that proprietary models exhibit lower robustness than open-sourced models under these perturbations.
>
> Moreover, lines 242–307 of the manuscript provide a detailed definition of CRS, which is composed of Confidence Change Rate (CCR), Average Confidence Change Magnitude (ACCM), and Significant Confidence Change Rate (SCCR), reflecting both the proportion and magnitude of confidence changes. The metric is straightforward and does not require further experiments to validate its effectiveness.

---

### Author Response · Authors · 2025-12-02
**Summary of Responses**

We sincerely thank the reviewers for their effort in reviewing our work and for their thoughtful comments. Below, we summarize the main strengths acknowledged by the reviewers and highlight the additional experiments we added in response to the reviews.

---

## Strengths Recognized by Reviewers

- Novel focus on step-level confidence reliability for MLLM-based Process Judges (MPJs) **(Reviewers C61o, Aftm, jaof)**.
- Three complementary metrics (CRS, CSS, CCS) covering robustness, sensitivity, and calibration **(Reviewers C61o, Aftm, jaof)**.
- Extensive experiments on both open-source and proprietary MLLMs **(Reviewers C61o, Aftm)**.
- Clear writing and systematic evaluation framework **(Reviewers C61o)**.

---

## Summary of Newly Added Experiments

To address reviewers' comments, we added several experiments:

- **Cross-perturbation robustness study (Responding to Reviewers Aftm, C61o):**
  Evaluated ConfProBench using perturbations generated by both GPT-4o and Gemini-2.5-Pro to verify that the benchmark is robust to different perturbation sources. Results show minimal score variations (<0.03) and stable relative rankings across MPJs.

- **Evaluation of latest MLLMs ( Responding to Reviewers Aftm):**
  Incorporated Gemini-2.5-Pro and GPT-5 into the benchmark. Key results include Gemini-2.5-Pro achieving new SOTA CRS (76.90) and CSS (57.73), and GPT-5 achieving second-best CSS (51.59) and CCS (55.38).

- **Human verification and inter-annotator agreement (Responding to Reviewers Aftm):**
  Two PhD-level annotators reviewed adversarial perturbations; rejection rate = 0.8%, percent agreement = 94.5%, Cohen’s Kappa = 0.812, confirming data quality.

These new updates have been incorporated into the final version of the paper.

---

### Note · Authors · 2026-01-02

I have read and agree with the venue's withdrawal policy on behalf of myself and my co-authors.